# Leveraging Knowledge of Modality Experts for Incomplete Multimodal Learning

## ABSTRACT

Multimodal Emotion Recognition (MER) may encounter incomplete multimodal scenarios caused by sensor damage or privacy protection in practical applications. Existing incomplete multimodal learning methods focus on learning better joint representations across modalities. However, our investigation shows that they are lacking in learning the unimodal representations which are rather discriminative as well. Instead, we propose a novel framework named Mixture of Modality Knowledge Experts (MoMKE) with two-stage training. In *unimodal expert training*, each expert learns the unimodal knowledge from the corresponding modality. In *experts mixing training*, both unimodal and joint representations are learned by leveraging the knowledge of all modality experts. In addition, we design a special Soft Router that can enrich the modality representations by dynamically mixing the unimodal representations and the joint representations. Various incomplete multimodal experiments on three benchmark datasets showcase the robust performance of MoMKE, especially on severely incomplete conditions. Visualization analysis further reveals the considerable value of unimodal and joint representations.

## CCS CONCEPTS

• **Information systems** → **Multimedia and multimodal retrieval**; Sentiment analysis; • **Computing methodologies** → **Natural language processing**.

## KEYWORDS

Incomplete multimodal learning, Multimodal emotion recognition, Modality knowledge expert, Soft router

## 1 INTRODUCTION

Multimodal learning [2, 31], which leverages complementary information across different modalities, has achieved significant strides in automatic emotion recognition [11, 27, 37]. Recently, Multimodal Emotion Recognition (MER) has become one of the most popular research topics in affective computing with many applications, including human-computer interactions [5, 9], dialogue systems [7, 13] and social media analysis [25].

Existing multimodal learning approaches in MER often implicitly assume that all modalities are available during training and testing. In the real world, however, factors such as sensor damage, automatic

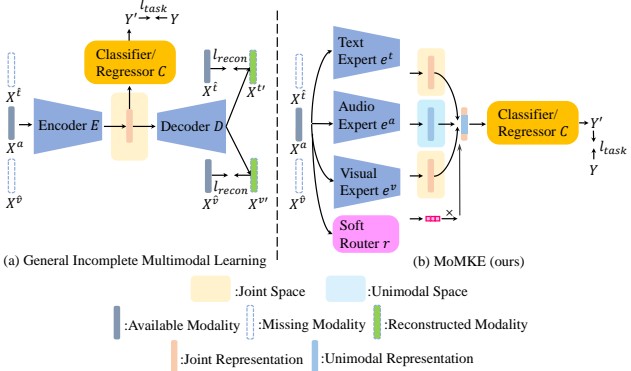

Figure 1: (a) **General Incomplete Multimodal Learning Methods focus on learning joint representations through the specifically designed encoder. The decoder ensures the quality of joint representations by reconstructing missing modalities during training. However, unimodal representations are ignored. (b) Our proposed MoMKE learns both unimodal and joint representations by leveraging the knowledge of all pre-trained modality experts.**

speech recognition errors, and privacy protection [36] often result in the unavailability of data from certain modalities. Recent studies [1, 18, 19] have shown that existing multimodal models trained on complete multimodal data are not inherently robust to incomplete multimodalities. Therefore, the development of a model capable of adapting to incomplete multimodalities constitutes a work of significant practical import at the present juncture.

The most intuitive approach to address incomplete multimodalities is to impute missing modalities, among which variational autoencoders (VAEs) [24, 30], generative adversarial networks (GANs) [4], and diffusion models [29] have been demonstrated to be capable of imputing missing modalities based on available ones. However, the quality of the imputed modalities highly depends on the architecture and parameters of the generative networks [14], and their high computational cost also limits their applications in scenarios requiring real-time processing [29]. Thus, an alternative approach, namely joint representation learning was widely accepted, which focuses on learning consistent joint representations across modalities. Essentially, all such methods follow a similar structural design, as shown in Fig. 1(a), where encoders encode available modalities into a joint space, while decoders constrain the learning of the joint representations through modality reconstruction only during training. Among these, Pham et al. [19] learned joint representations using a Seq2Seq model and cyclic translation loss. Zhao et al. [36] designed an encoder-decoder structure based on cascaded residual autoencoders and constrained joint representation learning through cycle consistency learning. Zuo et al. [39] further extracted

**Table 1: Performance of existing incomplete multimodal learning methods on the IEMOCAP dataset in severely incomplete (unimodal) conditions. Our proposed Mixture of Modality Knowledge Experts (MoMKE) significantly outperforms other methods.**

| Dataset | Models | Testing Condition | | |
|---|---|---|---|---|
| | | {a} | {t} | {v} |
| | | WA(%)/UA(%) | WA(%)/UA(%) | WA(%)/UA(%) |
| IEMOCAP | MCTN[19] | 49.75/51.62 | 62.42/63.78 | 48.92/45.73 |
| | MMIN[36] | **56.58/59.00** | 66.57/68.02 | 52.52/**51.60** |
| | IF-MMIN[39] | 55.03/53.20 | **67.02/68.20** | 51.97/50.41 |
| | MRAN[15] | 55.44/57.01 | 65.31/66.42 | **53.23**/49.80 |
| | **MoMKE(ours)** | **70.32/71.38** | **77.82/78.37** | **58.60/54.70** |
| | $\Delta_{SOTA}$ | ↑13.74/↑12.38 | ↑10.80/↑10.17 | ↑5.37/↑3.10 |

modality-invariant features based on the central moment discrepancy distance, while Luo et al. [15] aligned embeddings of other modalities around text-centricity.

Although these methods have achieved notable success to date, their performance is markedly suboptimal in severely incomplete conditions, i.e., only unimodal data, especially only visual or audio modality is available, as shown in Tab. 1. This phenomenon can be attributed to the neglect of learning unimodal representations, which in turn leads to insufficient discriminability in unimodal conditions.

Different from previous works, we explicitly define the unimodal and joint representations of the available modalities under incomplete multimodalities and propose the Mixture of Modality Knowledge Experts (MoMKE) with two-stage training to learn both of them, as illustrated in Fig. 1(b). Specifically, MoMKE first learns unimodal knowledge separately through *unimodal expert training*, enabling each expert to obtain discriminative unimodal representations. Then for each modality, MoMKE further learns the joint representations by leveraging the knowledge of other modality experts, and dynamically mixes them according to varied inputs through a proposed Soft Router to achieve more delicate representation fusion. We call this stage *experts mixing training*. In this way, MoMKE obtains more comprehensive representations for available modalities under incomplete multimodalities. Experiments on three MER datasets showcase the considerable effectiveness of MoMKE, especially, for severely incomplete conditions. Ablation studies and visualization analysis of expert loads further validate the significant value of unimodal representations and the necessity of joint representations under incomplete multimodalities. The main contributions of this paper can be summarized as follows:

1) We give the definitions of unimodal and joint representations for incomplete multimodalities, and based on this, propose the Mixture of Modality Knowledge Experts (MoMKE) to learn and mix them, surpassing the limitation of previous methods that only utilize joint representations.

2) To enrich the modality representations, MoMKE applies a two-stage training strategy that includes *unimodal expert training* and *experts mixing training* and utilizes a Soft Router to dynamically mix the unimodal and joint representations.

3) Experimental results on three benchmark datasets demonstrate that MoMKE outperforms existing state-of-the-art approaches

in incomplete multimodal learning. Specifically, in three severely incomplete conditions i.e., audio-only, text-only and visual-only, MoMKE improves average accuracy by 7.90%, 4.85%, and 4.55% compared to previous best methods, respectively.

## 2 RELATED WORK

### 2.1 Incomplete Multimodal Learning in MER

Learning from incomplete multimodalities is an important research topic in machine learning, with significant implications for applying models to complex real-world scenarios. One direct approach to address incomplete multimodalities is data imputation. Unsupervised imputation methods include zero imputation and average imputation [16, 18, 34]. Recently, deep learning based methods have utilized missing modalities as supervision to generate them through generative models such as VAE [24, 30] and GAN [4]. Tran et al. [26] proposed the cascaded residual autoencoder (CRA), which stacks residual autoencoders to simulate the correlations between different modalities, thereby imputing missing modalities from available ones. Wang et al. [29] leveraged a score-based diffusion model to map random noise to the distribution space of missing modalities, and used available modalities as semantic conditions to guide the denoising process for recovering missing modalities.

However, the high computational cost required by the imputation methods limits their application in real-time processing scenarios. Thereby methods that learn joint representations across modalities through consistency constraints are more widely used in incomplete multimodal learning for MER. Pham et al. [19] proposed the Multimodal Cyclic Translation Network (MCTN), learning joint representations through cyclic translation loss on Seq2Seq models. Zhao et al. [36] introduced the Missing Modality Imagination Network (MMIN) to learn multimodal joint representations through cross-modal imagination of missing modalities and cyclic consistency loss. Zuo et al. [39] further proposed IF-MMIN, which extracts invariant features between modalities based on central moment discrepancy distance. Luo et al. [15] proposed the Multimodal Reconstruction and Alignment Network (MRAN), enhancing model robustness by projecting visual and audio features into the text feature space. Lian et al. [12] proposed the Graph Completion Network (GCNet), utilizing the graph neural network for multimodal interaction to learn cross-modal representations. All these methods focused on learning joint representations across modalities through model design and constraints like reconstruction loss, but they overlooked the learning of the unimodal representations that contain modality-specific information.

In contrast to the aforementioned approaches, our method introduces unimodal expert training to learn unimodal representations, which are then effectively mixed with joint representations through a novel expert mixing training. This significantly enhances the discriminative power of the representations used for inferences.

### 2.2 Mixture-of-Expert (MoE)

In recent years, the Mixture of Experts (MoE) has been widely studied as a method to increase the model's capacity in parameter size without adding computational cost in natural language processing [6, 10, 22] and computer vision [17, 20, 23]. It typically consists of $N$ identical expert models and a router for assigning experts.

The router $r$ is a trainable gating function that assigns a score to each expert $e$ based on the input $X$. It then sparsely selects $K$ experts with top $K$ scores and uses a softmax function to compute the probability distribution of the outputs from the selected expert networks. Equation (1) mathematically represents MoE where $O$ is the output embeddings.

$$O = \sum_{i=1}^{K} \frac{Softmax(TopK(r(X), K))_i}{\sum_{j=1}^{K} Softmax(TopK(r(X), K))_j} e_i(X). \qquad (1)$$

In this way, MoE achieves specialization among different experts and sparsity in the model.

For the incomplete multimodal learning in MER, we focus on mixing the knowledge from all modality experts to enrich the modality representations instead of achieving sparsity of the model. Therefore, all modality experts are involved and routed, i.e., $K = N$, where each expert is pre-trained on the corresponding modality to learn unimodal knowledge. Hereinafter, this mechanism is referred to as the Soft Router, which is configured to dynamically mix the unimodal and joint representations under incomplete multimodalities. Its outputs (weights) indicate the importance of different representations for fusion.

## 3 METHOD

### 3.1 Preliminaries

In this subsection, we first formulate the problem of the General Incomplete Multimodal Learning in MER, then propose our new ideas to address the problem.

**Pre-define.** Following previous work [12], we consider incomplete multimodal learning in conversations, where audio $a$, text $t$ and visual $v$ modalities are utilized. Let's denote a conversation as $G = \{(u_i, y_i)\}_{i=1}^{L}$, where $L$ is the number of utterances in the conversation, $u_i$ is the $i^{th}$ utterance in $G$ and $y_i$ is the label of $u_i$. When under complete multimodalities, $X = \{X^m | m \in \{a, t, v\}\}$ is the multimodal feature set, where $X^m = \{x_i^m\}_{i=1}^{L}$ is the unimodal feature set. When under incomplete multimodalities, $X^M = \{X^m | m \in M\}$ denotes the feature set of available modalities, and $X^{\hat{M}} = \{X^{\hat{m}} | \hat{m} \in \hat{M}\}$ denotes the feature set of missing modalities, where $M \cup \hat{M} = \{a, t, v\}$ and $M \cap \hat{M} = \varnothing$. $Y = \{y_i\}_{i=1}^{L}$ denotes the label set. Please note that for general incomplete multimodal learning, all modalities are available during training. The missing modality $X^{\hat{M}}$ is not included in the inputs during testing, but can be used as supervision to train the model.

**General formulation.** Here, we summarize the existing general paradigm of incomplete multimodal learning in MER. As shown in Fig. 1(a), these methods typically consist of the following components: an encoder $E : X^M \rightarrow R$ that learns the mapping from available modalities to their representations, a classifier or regressor $C : R \rightarrow Y$ that maps the modality representations to labels, and a decoder $D : R \rightarrow X^{\hat{M}}$ that maps the modality representations to missing modalities. The optimization objective is:

$$min \quad l_{task}(C(E(X^M)), Y) + l_{recon}(D(E(X^M)), X^{\hat{M}}), \qquad (2)$$

where $l_{task}$ is the task loss to constrain the learning of task-related representations and $l_{recon}$ is the reconstruction loss that utilizes $X^{\hat{M}}$ as supervision to constrain the learned representations to lie

in the joint space. This enables the encoder $E$ to extract the joint representations that are shared between the available and missing modalities, i.e.,

$$E(X^M) = R_{joint}^M. \qquad (3)$$

Some methods may also impose additional loss constraints while their purpose remains to learn consistent joint representations across modalities.

However, our investigation shows that existing incomplete multimodal learning methods perform poorly in severely incomplete (unimodal) conditions. The possible reason is that the joint learning methods overlook the learning of the unimodal representations, that are not shared by other modalities, and the learned joint representations are less discriminative for the task. This raises a new question: how can one ensure the acquisition of a joint representation while simultaneously capturing unimodal representations under incomplete multimodalities?

To address this issue, we redefine the unimodal and joint representations in the framework of our approach.

**Proposed formulation.** In incomplete scenarios, there could be only one available modality, which is rather challenging to obtain joint representations through cross-modal interactions. Therefore, given a modality $m$, we distinguish its representations into unimodal and joint representations based on whether knowledge from other modalities is utilized during representation learning. Specially, given a unimodal feature set $X^m$, i.e. $M = \{m\}$, $m \in \{a, t, v\}$, we define the unimodal representation as $R_{uni}^m$ learned only from modality $m$, i.e.,

$$R_{uni}^m = e^M(X^m), \qquad (4)$$

where $e^M(\cdot)$ is an encoding function with the knowledge of modality $m$. We define the joint representation as $R_{joint}^m$ learned through the knowledge from other modalities $\hat{M}$, i.e.,

$$R_{joint}^m = e^{\hat{M}}(X^m), \qquad (5)$$

where $e^{\hat{M}}(\cdot)$ is an encoding function with the knowledge of other modalities $\hat{M}$. Due to the complexity of various incomplete multimodalities, modality representations of a unimodal feature $X^m$ become a mixed form. Thus for the $X^m$, its modality representations should be represented as

$$R^m = \alpha R_{uni}^m \oplus \beta R_{joint}^m = \alpha e^M(X^m) \oplus \beta e^{\hat{M}}(X^m), \qquad (6)$$

where $\alpha$ and $\beta$ are the weights of unimodal and joint representations determined by the inputs, and $\oplus$ denotes weighted fusion. Note that when the number of modalities exceeds two, $e^{\hat{M}}$ can be composed of multiple encoding functions with knowledge of different modalities. For example, in our scenario, when $m = a$, $R^a$ can be represented as

$$R^a = \alpha e^a(X^a) \oplus \beta_1 e^t(X^a) \oplus \beta_2 e^v(X^a), \qquad (7)$$

In this work, the objective of incomplete multimodal learning becomes to learn an encoder $E$ that encodes comprehensive modality representations $R^m$, i.e.,

$$E(X^M) = E(X^m) = R^m = \alpha R_{uni}^m \oplus \beta R_{joint}^m. \qquad (8)$$

Different from learning the encoder in Eq. (3), Eq. (8) also requires learning the unimodal representation and its weights with the joint representations to enrich the modality representations. In addition,

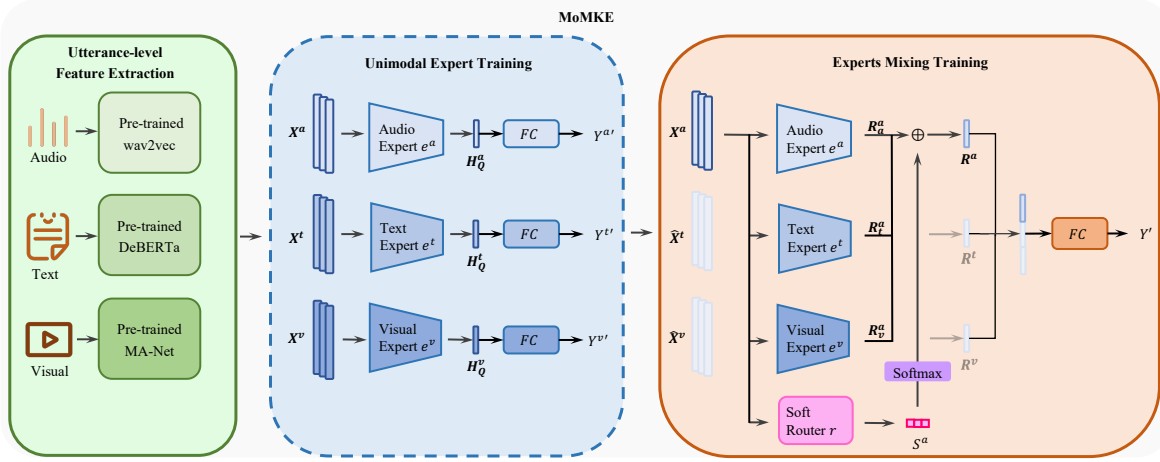

**Figure 2: The overall framework of MoMKE. During training, MoMKE conducts utterance-level feature extraction and extracts comprehensive representations under incomplete modalities through a two-stage training paradigm. In unimodal expert training (the blue block), each modality knowledge expert is separately trained using corresponding modality data and independent fully connected layers, respectively. In experts mixing training (the orange block), the unimodal and joint representations of each modality and their fusion weights (the output of Soft Router) are optimized based on all modality knowledge experts. The figure shows the audio-only condition, where the text and visual representations $R^t$ and $R^v$ (in light blue) are zero vectors. During testing, only the green block and orange block are applied.**

compared to learning unimodal representation only, the joint representations in Eq. (8) learned by leveraging knowledge from other modalities allow the model to achieve a more comprehensive understanding in unimodal conditions, which is validated in Section 4.4 and 4.6.

### 3.2 Overview Framework

To achieve Eq. (8), we proposed the Mixture of Modality Knowledge Experts (MoMKE) splitting the encoder into several modality knowledge experts and a Soft Router, i.e.,

$$E(X^m) = \sum_{k \in \{a,t,v\}} [r(X^m)]_k e^k(X^m), \qquad (9)$$

where $e^k(\cdot)$ is the expert with the knowledge of modality $k$, which is initialized by *unimodal expert training* on the corresponding modality. When $k = m$, $e^k(X^m) = R^m_{uni}$, otherwise $e^k(X^m) = R^m_{joint}$. $r(\cdot)$ is the fusion weights ($\alpha$, $\beta$ in Eq. (8)) obtained by the Soft Router learning). The optimization objective is:

$$min \quad l_{task}(C(E(X^m)), Y), \qquad (10)$$

$$i.e., \quad min \quad l_{task}(C(\sum_{k \in \{a,t,v\}} [r(X^m)]_k e^k(X^m)), Y). \qquad (11)$$

The overall framework of Mixture of Modality Knowledge Experts (MoMKE) is shown in Fig. 2. For each modality, utterance-level feature extraction is first performed. Then, the *unimodal expert training* (the blue block) trains each modality expert using its own modality data to learn unimodal knowledge. Afterward, in the most crucial *experts mixing training* (the orange blocks), all modality knowledge experts are trained to learn unimodal and joint representations, which will be further dynamically mixed by a Soft Router to obtain more comprehensive modality representations for various incomplete multimodal conditions.

### 3.3 Feature Extraction

For the audio modality of each utterance, the pre-trained wav2vec-large [21] model is used to extract 512-dimensional audio features. For the text modality, the pre-trained DeBERTa-large [8] model is utilized to encode each utterance into 1,024-dimensional text features. For the visual modality, the MTCNN [35] face detection algorithm is exploited to extract aligned faces, followed by facial feature extraction using the pre-trained MA-Net [38]. The frame-level features are compressed into 1,024-dimensional utterance-level features. Finally, for each utterance $u_i$, we extract multimodal features $x_i = \{x_i^m \in \mathbb{R}^{d_m} | m \in \{a, t, v\}\}$, where $x_i^a$, $x_i^t$ and $x_i^v$ are the respective utterance features of audio, text and visual modality and $d_a$, $d_t$ and $d_v$ are the feature dimensions corresponding to each modality.

### 3.4 Unimodal Expert Training

For each expert, we employ a transformer encoder structure. Given unimodal features $X^m = \{x_i^m\}_{i=1}^L \in \mathbb{R}^{L \times d_m}$, a projection matrix is applied to project them into a fixed dimension $d$:

$$Z^m = X^m W^m, \qquad (12)$$

where $W^m \in \mathbb{R}^{d_m \times d}$, $Z^m \in \mathbb{R}^{L \times d}$. Then, the position embeddings $Z^{pos}$ [28] are added to the features to encode the positional information of each utterance in the conversation:

$$H_0^m = Z^m + Z^{pos}, \qquad (13)$$

where $Z^{pos}$ and $H_0^m \in \mathbb{R}^{L \times d}$.

The encoded features will be fed into the modality transformer expert composed of $Q$ identical blocks, where each block consists of a multi-head self-attention (MSA) mechanism and feed-forward neural networks (FFN). Layer normalization (LN) and residual connections are applied within each block. The $j$-th transformer block

can be described as follows:

$$H_j^{m'} = MSA(LN(H_{j-1}^m)) + H_{j-1}^m$$
$$H_j^m = FFN(LN(H_j^{m'})) + H_j^{m'}, \tag{14}$$

The final output $H_Q^m$ of the modality expert transformer will be processed through independent fully-connected layers to obtain labels $Y^{m'} = \{y^{m'}\}_{i=1}^L$,

$$H_Q^m = Transformer_{\theta_T^m}(X^m),$$
$$Y^{m'} = FC_{\theta_{FC}^m}(H_Q^m), \tag{15}$$

where $H_Q^m \in \mathbb{R}^{L \times d}$, $\theta_T^m$ and $\theta_{FC}^m$ represent the parameters of the transformer expert and fully-connected layers, respectively. The transformer expert is optimized by minimizing the task-specific loss, such as cross-entropy loss

$$l_{task} = CrossEntropy(Y, Y^{m'}), \tag{16}$$

for emotion recognition, or mean square error (MSE) loss, denoted as

$$l_{task} = MSE(Y, Y^{m'}), \tag{17}$$

is used for sentiment analysis.

After unimodal expert training, each modality expert has learned the unimodal knowledge, equivalent to the expert $e^m(\cdot)$ in Eq. (9), i.e., $Transformer_{\theta_T^m}(\cdot) = e^m(\cdot)(m \in \{a, t, v\})$.

## 3.5 Experts Mixing Training

After unimodal expert training, each modality expert has learned the ability of extracting unimodal representations. At this stage, our objective is to learn both unimodal and joint representations based on all modality knowledge experts and dynamically mix them to better cope with various incomplete multimodal conditions. Taking the case where only audio modality is available as an example, i.e. $m = a$, the audio features will be processed by both the audio expert and the experts with knowledge of other modalities to obtain the unimodal and joint representations of audio:

$$R_a^a = e^a(X^a) = Transformer_{\theta_T^a}(X^a),$$
$$R_t^a = e^t(X^a) = Transformer_{\theta_T^t}(X^a), \tag{18}$$
$$R_v^a = e^v(X^a) = Transformer_{\theta_T^v}(X^a),$$

where $R_a^a$ is the audio unimodal representations corresponding to $R_{uni}^m$ in Eq. (8) ($m = a$), $R_t^a$ and $R_v^a$ are the joint representations of audio features incorporating text and visual knowledge corresponding to $R_{joint}^a$ in Eq. (8) ($m = a$). It is analogous when text or visual modality is available.

Due to the complexity of incomplete multimodalities, such as varied data qualities, tasks, and so on, the unimodal representations and joint representations may largely change. A fixed fusion strategy (such as a simple weighted average) can hardly achieve optimal results in all cases. Therefore, a Soft Router $r$, i.e., a two-layer Multilayer Perceptron (MLP), is further introduced to dynamically measure the importance (weights) of the unimodal and joint representations according to different inputs, which will be used to mix the unimodal and joint representations to obtain the audio representation $R^a$:

$$S^a = [s_a^a, s_t^a, s_v^a] = r(X^a) = MLP(X^a), \tag{19}$$

$$w_i^a = softmax(s_i^a) = \frac{e^{s_i^a}}{\sum_{j \in \{a,t,v\}} e^{s_j^a}}, \tag{20}$$

$$R^a = \sum_{i \in \{a,t,v\}} w_i^a \cdot R_i^a, \tag{21}$$

where $S^a \in \mathbb{R}^{L \times 3}$, $R^a \in \mathbb{R}^{L \times d}$, $w_i^a$ is the weights corresponding to $\alpha$ and $\beta$ in Eq. (8) and $R^a$ corresponds to the modality representations $R^m$. By observing how the Soft Router dynamically adjusts the weight of the representations, deeper insights into the roles of unimodal and joint representations can be gained. This is evaluated in the Section 4.5.

The audio representation $R^a$, which mixes knowledge from all modality experts, is fed into a fully connected layer to obtain labels:

$$Y^{m'} = FC_{\theta_{FC'}^m}(R^m). \tag{22}$$

The optimization objective is given by Eq. (11), where the loss is either Eq. (16) or Eq. (17). When more modalities are available, the features of each modality still follow the above forward process to obtain its representation $R^m$. All modality representations are concatenated in the feature dimension to obtain a fused representation before inference. During the training process, the Soft Router and all the experts are trainable to learn the corresponding knowledge for extracting and mixing modality representations in various incomplete multimodal conditions.

# 4 EXPERIMENT

## 4.1 Datasets and Evaluation Metrics

To assess the effectiveness of MoMKE, we conduct experiments on three benchmark multimodal emotion recognition or sentiment analysis datasets, including IEMOCAP, CMU-MOSI, and CMU-MOSEI.

**IEMOCAP** dataset [3] consists of 5 dyadic dialogue sessions, where actors perform improvisations or scripted scenarios. Each dialogue is further segmented into numerous utterances, with each utterance annotated with categorical emotion labels. For a fair comparison, we follow previous works [12, 36, 39] to form the four-class emotion (happy, sad, neutral and angry) recognition setup.

**CMU-MOSI** dataset [32] is a collection of 2199 opinionated video clips collected from YouTube, where 1284, 229, and 686 of them are used as training, validation, and test set. Each video clip is labeled with a sentiment score ranging from -3 (strongly negative) to +3 (strongly positive).

**CMU-MOSEI** dataset [33] contains 22856 video clips from over 1000 online YouTube speakers, where 16326 of them are used for training, the remaining 1871 and 4659 of them are used for validation and testing. All utterances are randomly selected from a variety of topic and monologue videos and follow the annotation scheme of [-3, 3] in CMU-MOSI.

For the IEMOCAP dataset, we adopt weighted accuracy (WA) and unweighted accuracy (UA) as evaluation metrics. For the CMU-MOSI and CMU-MOSEI datasets, we focus on negative/positive classification tasks in line with previous works [12, 15, 29] where negative and positive classes are assigned for < 0 and > 0 sentiment scores, and utilize ACC and F1 scores as evaluation metrics.

Table 2: Comparison with state-of-the-art (SOTA) methods on three benchmark datasets under all possible incomplete testing conditions (e.g. testing case {$a$} means that only audio modality is available and both text and visual modalities are missing). "Average" refers to the average performance over all six incomplete multimodal conditions. The overall best results of each dataset are highlighted in red bold, while the second-best results are in black bold. The results with $^a$ are from [29], and the others can be found in their paper. The row with $\Delta_{SOTA}$ means the improvements or reductions of MoMKE compared to the best competing methods.

| Datasets | Models | Testing Condition | | | | | | | |
|---|---|---|---|---|---|---|---|---|---|
| | | {$a$} | {$t$} | {$v$} | {$a,v$} | {$a,t$} | {$t,v$} | Average | {$a,t,v$} |
| | | WA(%)/UA(%) | WA(%)/UA(%) | WA(%)/UA(%) | WA(%)/UA(%) | WA(%)/UA(%) | WA(%)/UA(%) | WA(%)/UA(%) | WA(%)/UA(%) |
| IEMOCAP | MCTN[19] | 49.75/51.62 | 62.42/63.78 | 48.92/45.73 | 56.34/55.84 | 68.34/69.46 | 67.84/68.34 | 58.94/59.13 | - |
| | MMIN[36] | **56.58**/**59.00** | 66.57/68.02 | 52.52/**51.60** | 63.99/65.43 | 72.94/75.14 | 72.67/73.61 | 64.10/65.24 | - |
| | IF-MMIN[39] | 55.03/53.20 | **67.02**/**68.20** | 51.97/50.41 | **65.33**/**66.52** | **74.05**/**75.44** | **72.68**/**73.62** | **64.54**/**65.38** | - |
| | MRAN[15] | 55.44/57.01 | 65.31/66.42 | **53.23**/49.80 | 64.70/64.46 | 73.00/74.58 | 72.11/72.24 | 63.97/64.08 | - |
| | **MoMKE(ours)** | 70.32/71.38 | 77.82/78.37 | 58.60/54.70 | 68.85/67.65 | 79.89/79.53 | 77.87/77.84 | 72.23/71.58 | 80.13/79.99 |
| | $\Delta_{SOTA}$ | ↑13.74/↑12.38 | ↑10.80/↑10.17 | ↑5.37/↑3.10 | ↑3.52/↑1.13 | ↑5.84/↑4.09 | ↑5.19/↑4.22 | ↑7.69/↑6.20 | - |
| | | ACC(%)/F1(%) | ACC(%)/F1(%) | ACC(%)/F1(%) | ACC(%)/F1(%) | ACC(%)/F1(%) | ACC(%)/F1(%) | ACC(%)/F1(%) | ACC(%)/F1(%) |
| CMU-MOSI | MCTN$^a$[19] | 56.10/54.50 | 79.10/79.20 | 55.00/54.40 | 57.50/57.40 | 81.00/81.00 | 81.10/81.20 | 68.30/67.95 | 81.40/81.50 |
| | MMIN$^a$[36] | 55.30/51.50 | 83.80/83.80 | 57.00/58.50 | 60.40/58.50 | 84.00/84.00 | 83.80/83.90 | 70.72/69.28 | 84.60/84.40 |
| | GCNet$^a$[12] | 56.10/54.50 | 83.70/83.60 | 56.10/55.70 | 62.00/61.90 | 84.50/84.40 | 84.30/84.20 | 71.12/70.72 | 85.20/85.10 |
| | IMDer$^a$[29] | **62.00**/**62.20** | **84.80**/**84.70** | **61.30**/**60.80** | **63.60**/**63.40** | **85.40**/**85.30** | **85.50**/**85.40** | **73.77**/**73.63** | **85.70**/**85.60** |
| | **MoMKE(ours)** | 63.19/58.61 | 86.59/86.52 | 63.35/63.34 | 64.04/64.66 | 87.20/87.17 | 87.04/87.00 | 75.24/74.55 | 87.96/87.89 |
| | $\Delta_{SOTA}$ | ↑1.19/↓3.59 | ↑1.79/↑1.82 | ↑2.05/↑2.54 | ↑0.44/↑1.26 | ↑1.80/↑1.87 | ↑1.54/↑1.60 | ↑1.47/↑0.92 | ↑2.26/↑2.39 |
| CMU-MOSEI | MCTN$^a$[19] | 62.70/54.50 | 82.60/82.80 | 62.60/57.10 | 63.70/62.70 | 83.50/83.30 | 83.20/83.20 | 73.05/70.60 | 84.20/84.20 |
| | MMIN$^a$[36] | 58.90/59.50 | 82.30/82.40 | 59.30/60.00 | 63.50/61.90 | 83.70/83.30 | 83.80/83.40 | 71.92/71.75 | 84.30/84.20 |
| | GCNet$^a$[12] | 60.20/60.30 | 83.00/83.20 | 61.90/61.60 | 64.10/61.60 | 84.30/84.40 | 84.30/84.40 | 73.10/72.80 | **85.20**/**85.10** |
| | IMDer$^a$[29] | **63.80**/**60.60** | **84.50**/**84.50** | **63.90**/**63.60** | **64.90**/**63.50** | **85.10**/**85.10** | **85.00**/**85.00** | **76.00**/**75.30** | 85.10/**85.10** |
| | **MoMKE(ours)** | 72.56/71.03 | 86.46/86.43 | 70.12/70.23 | 73.34/71.82 | 86.68/86.61 | 86.79/86.69 | 79.33/78.80 | 87.12/87.03 |
| | $\Delta_{SOTA}$ | ↑8.76/↑10.43 | ↑1.96/↑1.93 | ↑6.22/↑6.63 | ↑8.44/↑8.32 | ↑1.58/↑1.51 | ↑1.79/↑1.69 | ↑4.80/↑5.08 | ↑2.02/↑1.93 |

## 4.2 Implementation Details

Following previous works [15, 29, 36, 39], we investigate the performance under fixed missing protocol, where certain modalities are completely missing during testing. There are a total of 6 incomplete multimodal conditions, i.e., {$a$}, {$t$}, {$v$}, {$a,v$}, {$a,t$} and {$t,v$}. We also evaluate under the complete multimodal conditions, i.e., {$a,t,v$}. For the IEMOCAP, CMU-MOSI, and CMU-MOSEI datasets, the dimensions $d$ of the modality representations are set to 256, 128, and 256, respectively, and the maximum training epochs for both the pre-training and boosting stage are set to 100, 100, and 50, respectively. The number of blocks $Q$ in the transformer and the number of heads in the multi-head self-attention are set to 4 and 2, respectively. We use the Adam optimization with a learning rate of 0.0001, and the dropout rate is set to 0.5 for all datasets. For IEMOCAP, we perform five-fold cross-validation using the leave-one-session-out strategy. For CMU-MOSI and CMU-MOSEI, we run each experiment five times and report the average results.

## 4.3 Comparison with SOTA methods

To thoroughly evaluate the performance of MoMKE under various incomplete multimodal conditions, we compare it on three benchmark datasets against state-of-the-art (SOTA) methods that utilize identical fixed missing protocol, including MCTN[19], MMIN[36], IF-MMIN[39], MRAN[15], GCNet[12] and IMDer[29]. As shown in Tab. 2, MoMKE achieves the highest performance in almost all evaluation metrics under various incomplete multimodalities across all datasets. It can be observed that competing methods perform much worse under severely incomplete multimodalities (i.e. unimodal conditions {$a$}, {$t$}, and {$v$}), especially when only visual or audio

modality is available. In contrast, MoMKE significantly mitigates this phenomenon, showing notable improvements over competing methods, except for the {$a$} case of CMU-MOSI, where the F1 metric is sub-optimal. Specifically, under the {$a$}, {$t$} and {$v$} testing conditions, MoMKE improved the WA/ACC by an average of 7.90%, 4.85%, and 4.55% on three datasets, respectively. This can be attributed to MoMKE's ability to enhance the modality representations of the unimodal data by mixing the joint representations with the unimodal representations. In other incomplete multimodal conditions ({$a,v$}, {$a,t$} and {$t,v$}), MoMKE also outperforms other methods. This can be explained by MoMKE learning more comprehensive modality representations, thereby improving the post-fusion performance. Overall, on IEMOCAP, MoMKE outperforms the best competing method by an average of 7.69% and 6.2% on WA and UA. On CMU-MOSEI, MoMKE outperforms the best competing methods by an average of 4.80% and 5.08% on ACC and F1. However, MoMKE only achieves 1.47% and 0.92% improvements on CMU-MOSI. The possible reason could be the small scale of the dataset, which makes the model prone to overfitting. Moreover, it is very interesting that MoMKE also demonstrates superiority over all competing methods under complete multimodalities.

## 4.4 Ablation Study

In this subsection, we conduct ablation studies to gain deeper insights into the roles played by each configuration and expert in MoMKE.

**Ablation of the training stages and the router:** To explore the effectiveness of the two training stages and the Soft Router in MoMKE, we conduct the following ablation experiments:

**Table 3: Ablation results of the training stages and the router.**

| Datasets | Modules | {a} | {t} | {v} | {a,v} | {a,t} | {t,v} | Average | {a,t,v} |
|---|---|---|---|---|---|---|---|---|---|
| | | Testing Condition | | | | | | | |
| | | WA(%)/UA(%) | WA(%)/UA(%) | WA(%)/UA(%) | WA(%)/UA(%) | WA(%)/UA(%) | WA(%)/UA(%) | WA(%)/UA(%) | WA(%)/UA(%) |
| IEMOCAP | Without unimodal expert training | 67.89/68.56 | 75.30/76.54 | 56.97/53.09 | 66.58/65.91 | 77.36/77.93 | 76.99/76.81 | 70.02/69.64 | 77.62/77.47 |
| | Without experts mixing training | 67.88/67.91 | 75.74/75.34 | 56.87/53.37 | 66.65/65.46 | 77.25/78.33 | 76.99/76.93 | 70.06/69.39 | 77.17/77.80 |
| | Without router | 68.32/69.03 | 76.22/76.30 | 57.00/53.70 | 67.10/66.32 | 78.78/78.53 | 77.00/77.21 | 70.74/70.18 | 78.78/78.23 |
| | **MoMKE(ours)** | **70.32/71.38** | **77.82/78.37** | **58.60/54.70** | **68.85/67.65** | **79.89/79.53** | **77.87/77.84** | **72.23/71.58** | **80.13/79.99** |
| | | ACC(%)/F1(%) | ACC(%)/F1(%) | ACC(%)/F1(%) | ACC(%)/F1(%) | ACC(%)/F1(%) | ACC(%)/F1(%) | ACC(%)/F1(%) | ACC(%)/F1(%) |
| CMU-MOSI | Without unimodal expert training | 56.09/56.36 | 85.20/85.09 | 62.20/61.82 | 63.41/63.57 | 86.20/86.13 | 85.50/85.51 | 73.10/73.08 | 86.89/86.85 |
| | Without experts mixing training | 59.76/57.34 | 85.04/85.94 | 62.35/61.96 | 63.10/63.28 | 86.04/85.64 | 85.35/85.34 | 73.61/73.25 | 87.20/87.08 |
| | Without router | 61.31/58.56 | 85.30/86.10 | 62.20/61.82 | 63.58/63.20 | 86.20/86.43 | 85.78/85.71 | 74.06/73.64 | 87.00/86.70 |
| | **MoMKE(ours)** | **63.19/58.61** | **86.59/86.52** | **63.35/63.34** | **64.04/64.66** | **87.20/87.17** | **87.04/87.00** | **75.24/74.55** | **87.96/87.89** |
| CMU-MOSEI | Without unimodal expert training | 71.18/70.13 | 84.93/84.95 | 68.23/67.39 | 71.18/70.52 | 84.23/84.24 | 85.26/85.25 | 77.50/77.08 | 86.20/86.06 |
| | Without experts mixing training | 70.85/69.37 | 85.01/85.84 | 68.23/67.08 | 70.40/70.12 | 84.20/84.10 | 85.34/85.25 | 77.17/76.79 | 86.51/86.43 |
| | Without router | 71.59/70.46 | 85.45/85.48 | 68.99/68.00 | 72.00/71.18 | 84.88/85.12 | 85.34/85.55 | 78.04/77.63 | 86.89/86.43 |
| | **MoMKE(ours)** | **72.56/71.03** | **86.46/86.43** | **70.12/70.23** | **73.34/71.82** | **86.68/86.61** | **86.79/86.69** | **79.33/78.80** | **87.12/87.03** |

1) *Without unimodal expert training*: The model structure remains the same as MoMKE, but the three transformers are randomly initialized instead of modality knowledge experts.

2) *Without experts mixing training*: Modality knowledge experts corresponding to missing modalities will not be used for representations mixing, i.e., only the modality experts corresponding to the available modalities are utilized. In this setting, for the unimodal conditions, the model is the corresponding modality transformer expert, while for the multimodal conditions, the inference is obtained by averaging the outputs of each available-modality transformer expert.

3) *Without router*: The model structure is the same as MoMKE, but without the Soft Router, i.e., the weights of all experts are equal and the modality representation is the sum-average of all representations.

The results of the ablation experiments are shown in Tab. 3. When unimodal expert training is removed, the performance drops significantly across all metrics. This highlights the importance of building unimodal knowledge experts, which can help to learn discriminative unimodal representations and further aid joint representation learning. A similar degradation can be observed when experts mixing training is removed, which demonstrates that the knowledge from other modality experts can help to learn more comprehensive representations, thereby improving the performance of MoMKE. Compared with the above two ablations, the performance of the model in *Without router* is better since it can learn both the unimodal and joint representations. However, there is still a significant gap compared to MoMKE, which suggests that the contributions of unimodal representations and joint representations inherently vary in diverse incomplete scenarios. Nonetheless, a well-trained Soft Router is able to dynamically adjust the weights to mix these representations according to the particularities of the incomplete scenario, thereby enhancing the modality representations.

**Ablation of the experts:** To explore the impact of each modality expert on learning modality representations, we conduct experiments using different combinations of modality experts during experts mixing training. Table 4 shows the experimental results in the severely incomplete (unimodal) conditions on IEMOCAP. We divide the experiment settings into two groups: one in which the expert corresponding to the available modality is used in MoMKE

**Table 4: Results of the experts selected by MoMKE during experts mixing training in the severely incomplete (unimodal) conditions ({a},{t},{v}) on IEMOCAP. "Expert" represents the modality experts used during experts mixing training. The underlined results represent the experiments using the expert corresponding to the available modality.**

| Expert | {a} | {t} | {v} |
|---|---|---|---|
| | Testing Condition | | |
| | WA(%)/UA(%) | WA(%)/UA(%) | WA(%)/UA(%) |
| a | 67.88/67.91 | 73.37/72.12 | 54.48/52.40 |
| t | 65.00/65.54 | 75.74/75.34 | 54.98/53.00 |
| v | 64.90/64.44 | 72.37/73.13 | 56.87/53.37 |
| a,v | 68.70/69.44 | 74.34/74.03 | 57.33/53.51 |
| a,t | 68.38/69.27 | 76.65/76.84 | 55.48/53.48 |
| t,v | 66.00/66.87 | 76.96/76.55 | 57.25/53.97 |
| a,t,v | **70.32/71.38** | **77.82/78.37** | **58.60/54.70** |

(with underline), and the other in which the corresponding expert is not used (without underline). Results under each testing condition, i.e., in each column, show a clear conclusion that the models using the expert of the corresponding modality outperform those that do not use it. This validates our hypothesis that unimodal representations play a more crucial role in severely incomplete conditions. Furthermore, it can be observed that, based on utilizing unimodal representations, mixing more experts can achieve better performance. This highlights the importance of joint representations and shows that mixing the knowledge of more modality experts can help to obtain a more comprehensive modality representation.

## 4.5 Visualization Analysis

To have an insight into the role of unimodal and joint representations in severely incomplete conditions, we visualize the expert loads of the test set during experts mixing training, i.e., the weight of each modality expert assigned by the Soft Router. As shown in Fig. 3, as training progresses, the model gradually converges, and the expert loads stabilize. The visualization results indicate that the model tends to rely more on the corresponding unimodal representations, which contain more modality-specific discriminative information, again being in line with our hypothesis. At the same

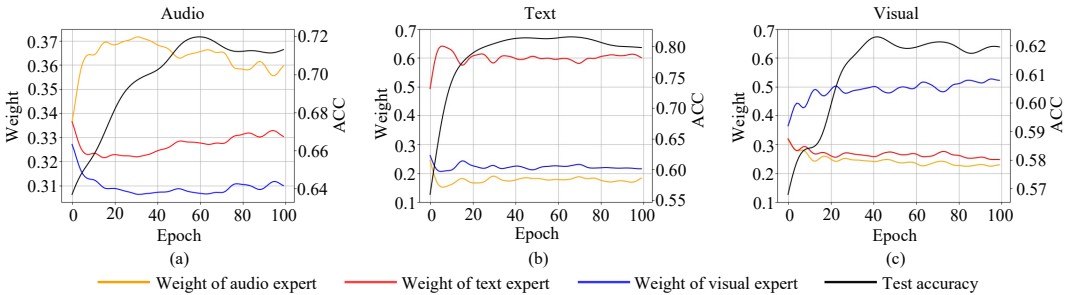

Figure 3: Trends in expert loads and test accuracy during experts mixing training on IEMOCAP in the severely incomplete (unimodal) conditions. Each subplot represents a severely incomplete case. The horizontal axis denotes the training epochs, the left vertical axis is the expert loads after softmax, which are calculated by averaging the expert loads of all test samples, and the right vertical axis is the test accuracy.

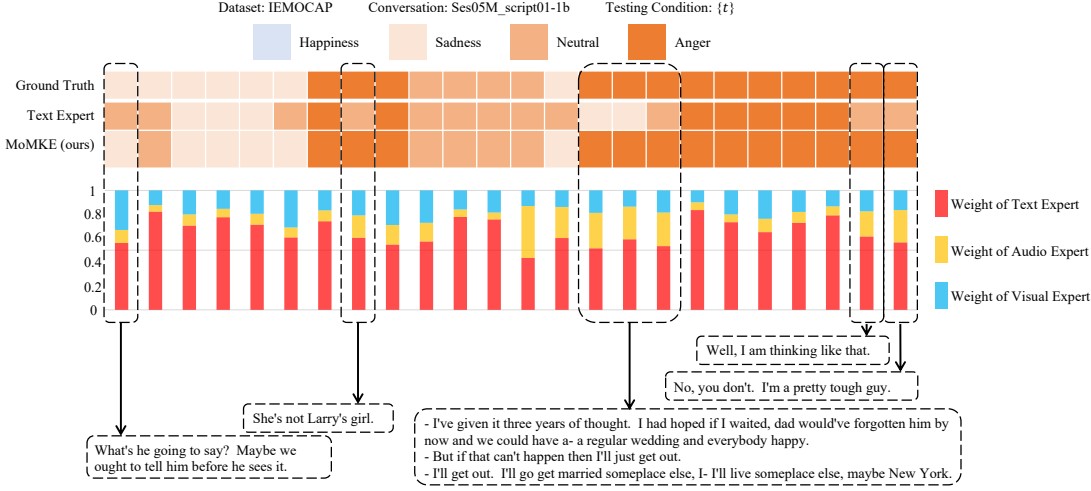

Figure 4: The inference of MoMKE and Text Expert (only the text expert is used) on a conversation in the text-only case {$t$}, along with the expert loads in MoMKE. This conversation comes from *Ses05M_script01_1b* of the IEMOCAP dataset, where the speakers' emotions progress from sadness to neutral to anger.

time, the joint representations also account for a certain portion of the modality representations, which demonstrates their necessity.

## 4.6 Case Study

To further illustrate the necessity of learning joint representations through other modality knowledge experts, we carry out a case study by selecting a conversation from the IEMOCAP dataset, then visualize the expert loads in MoMKE when only text modality is available, and compare the inferences of using only text expert and MoMKE, as shown in Fig. 4. It can be observed that when the model relies solely on text expert with the text unimodal representations, the inferences tend to be neutral or sadness, as indicated by the black dashed box. In contrast, MoMKE can leverage the knowledge from all modality experts to do correct inferences by mixing more joint representations from other modality experts, imagining the speakers' tone and expression from the text as a human would, thus obtaining a more comprehensive modality representation.

## 5 CONCLUSION

In this paper, we propose a novel framework named Mixture of Modality Knowledge Experts (MoMKE) with two-stage training to enrich the modality representations under incomplete multi-modalities in MER. MoMKE goes beyond the limitations of existing works learning insufficient unimodal representation by leveraging unimodal experts, and takes a step further to learn joint representations through other modality experts and dynamically mix them via a Soft Router. Experiments demonstrate the robustness of MoMKE to adapt to various incomplete multimodal scenarios, with ablation studies and visualization analysis further confirming that this capability stems from the mixture of unimodal and joint representations.

Currently, MoMKE still assumes that all modality data are available during training, However, it may not possible to obtain complete modality training data in real-world scenarios. In future work, we will explore modality representation learning in the context of incomplete multimodalities for model training.

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
