# OpenReview forum: "Leveraging Knowledge of Modality Experts for Incomplete Multimodal Learning"
_acmmm.org/ACMMM/2024/Conference — MM2024 Oral_

### Official Review · Reviewer_dpcL · 2024-05-14

**Rating:** 5
**Confidence:** 3

**Summary:**

This research centers on the Multimodal Emotion Recognition (MER) task, particularly addressing Incomplete Multimodal Learning within MER. The study introduces the Mixture of Modality Knowledge Experts (MoMKE) to both learn and blend modalities, overcoming the constraint of prior methods that solely rely on joint representations. Additionally, it presents a two-stage training approach involving unimodal expert training and expert mixing training, employing a Soft Router to combine the unimodal and joint representations dynamically.

**Strengths:**

1. The proposed idea is novel and good, choosing an important problem to focus on and solve.

2. The paper is written well.

3. The experimental setup and analysis are good.

**Limitations:**

1. No mention of the number of seeds used to finetune the models. It is not clear if average and standard deviation are being reported for the tests. Generally, one must evaluate multiple seeds and then report the average and standard deviation.

2. Previous works in MSA running on CMU-MOSI and CMU-MOSEI reported MAE as a main metric since these tasks are considered regression tasks (labeled from -3 to 3). You should consider adding these metrics and compare them with some other methods.

3. I can not find \hat{X}t and \hat{X}v in Experts Mixing Training in Figure 2.

**Suitability:**

2

---

### Official Review · Reviewer_LcNG · 2024-05-15

**Rating:** 6
**Confidence:** 3

**Summary:**

Multimodal Emotion Recognition (MER) may encounter incomplete multimodal scenarios caused by sensor damage or privacy protection in practical applications. Existing incomplete multimodal learning methods focus on learning better joint representations across modalities. However, author's investigation shows they are lacking in learning the unimodal representations which are rather discriminative as well. This paper proposes a framework for MER task.

**Strengths:**

This paper proposes a novel framework Mixture of Modality Knowledge Experts (MoMKE) with two-stage training. In the unimodal expert training, each expert learns the knowledge of the unimodal from the corresponding modality. In experts mixing training, both unimodal and joint representations are learned by leveraging the knowledge of all modality experts. The experimental results increase greatly.

**Limitations:**

(1) The proposed two-stage training model will inevitably increase a large number of model parameters and computational cost, whether you can give the changes of the number of parameters and the impact on the running time.
(2) In the case of missing modalities (e.g., only audio is available), whether the audio modality extracted and jointly represented by the unimodal expert of other modalities will have a noise effect on the features of the original audio processed by the unimodal expert.
(3) How the unimodal joint representation of the model will be handled if only one modality is missing. It is suggested that a relevant explanation be added to this situation.

**Suitability:**

3

---

### Official Review · Reviewer_GvnS · 2024-05-24

**Rating:** 5
**Confidence:** 3

**Summary:**

The paper presents a novel framework named Mixture of Modality Knowledge Experts (MoMKE) designed to address the issue of incomplete multimodal scenarios in Multimodal Emotion Recognition (MER). The proposed approach involves a two-stage training strategy: unimodal expert training and experts mixing training. This framework aims to enrich modality representations by leveraging both unimodal and joint representations through a Soft Router mechanism. Experimental results on three benchmark datasets (IMEOCAP, CMU-MOSI, CMU-MOSEI) demonstrate the robustness and effectiveness of MoMKE, particularly under severely incomplete conditions.

**Strengths:**

Novelty: The introduction of the Mixture of Modality Knowledge Experts (MoMKE) framework is a novel approach to handling incomplete multimodal learning. The two-stage training strategy effectively addresses the limitations of existing methods that focus solely on joint representations.

Technical Correctness: The framework is well-justified with a solid theoretical foundation. The use of a Soft Router to dynamically mix unimodal and joint representations is technically sound and innovative. The ablation study in Table 3 presents the effectiveness of the designed modules, especially the Soft Roter.

Evaluation: The experiments are thorough and cover various incomplete multimodal scenarios. The results showcase significant improvements over state-of-the-art methods, particularly in severely incomplete conditions, as shown in Table 2.

Clarity: The paper is well-written and structured, making the concepts and methodologies clear and easy to follow. The inclusion of visualization analysis further aids in understanding the impact of the proposed approach.

**Limitations:**

Assumption on Training Data: Like previous works, the framework in this work still assumes that all modality data are available during training, which might not always be practical in real-world scenarios. Future work could explore scenarios with incomplete modality training data.

Unfair comparison: This paper proposes to use pre-trained wav2veclarge, pre-trained DeBERTa-large and pre-trained MA-Net for audio, text and visual features, respectively. These pre-trained models are not used in other works at the same time. The improvement in this paper could be mainly introduced by the more powerful unimodal encoders instead of the proposed framework. The author could try to produce some results with the common unimodal encoders, or no pretrained encoders.

**Suitability:**

3

---

### Official Review · Reviewer_1ftq · 2024-05-25

**Rating:** 4
**Confidence:** 3

**Summary:**

The paper proposed a novel incomplete multimodal learning framework, named MoMKE, to solve the problem of lacking in learning the unimodal representations due to focusing on learning better joint representation. The proposed framework consists of two-stage training, which is unimodal expert training and experts mixing training with Soft Router to dynamically "mix" the unimodal representations and joint representations. The effects of the proposed framework are verified with three popular multimodal sentiment analysis or emotion recognition benchmarks, IEMOCAP, CMU-MOSI, and CMU-MOSEI, as well as ablation analysis for demonstration of each module’s effectiveness.

**Strengths:**

[S1] The related works are well-structured to understand the limitations of previous multimodal fusion works in incomplete conditions, as well as their contributions compared to previous works.


[S2] They carefully defined unimodal, joint representation and their proposed scenarios using dedicated formulations.


[S3] The ablation study is well-designed, especially for the case study in this paper, which effectively demonstrates why mixing the modality experts is necessary for multimodal emotion recognition.

**Limitations:**

[W1] Technical contribution is unclear. It is a low contribution if the proposed framework is just similar to the combination of the MoE-based fusion model[1] and SMEAR[2] that use a mixture of experts for the fusion of each modality and dynamic merging of experts with adaptive routing, respectively. The paper needs to describe the key differences from the naive solution that combines existing methods.


[W2] The baseline is weak to validate the performance of the proposed model. MTMSA[3] outperforms state-of-the-art methods, and the proposed framework at the IEMOCAP benchmark, and MTMSA reported performance trends according to the missing modality rate.


[W3] Writing needs improvement. Section 3.1 is overly detailed, especially lines 298-306, which unnecessarily repeat information from the introduction.

[1] Goyal, Ankit, et al. "A multimodal mixture-of-experts model for dynamic emotion prediction in movies." 2016 ieee international conference on acoustics, speech and signal processing (ICASSP). IEEE, 2016.
[2] Muqeeth, Mohammed, Haokun Liu, and Colin Raffel. "Soft merging of experts with adaptive routing." arXiv preprint arXiv:2306.03745 (2023).
[3] Liu, Zhizhong, et al. "Modality translation-based multimodal sentiment analysis under uncertain missing modalities." Information Fusion 101 (2024): 101973.

**Suitability:**

3

---

### Meta-Review · Area_Chair_A4yt · 2024-07-10

**Recommendation:** Accept (Oral)
**Confidence:** 5

**Metareview:**

All the reviewers gave positive ratings and tend to accept the paper. SAC and AC agree with reviewers and recommend acceptance of the paper.